# Scrambling the skin: A psychophysical study of adaptation to scrambled tactile apparent motion

**Tatjana Seizova-Cajic**[1]*, **Sandra Ludvigsson**[2●], **Birger Sourander**[2●], **Melinda Popov**[2], **Janet L. Taylor**[3,4]

**1** Faculty of Medicine and Health, University of Sydney, Sydney, Australia, **2** Linkoping University, Linkoping, Sweden, **3** Neuroscience Research Australia, Sydney, Australia, **4** School of Medical and Health Sciences, Edith Cowan University, Perth, Australia

● These authors contributed equally to this work.

* tatjana.seizova-cajic@sydney.edu.au

**Data Availability Statement:** Data files are available from the Open Science Framework database. DOI 10.17605/OSF.IO/GTCR7.

## Abstract

An age-old hypothesis proposes that object motion across the receptor surface organizes sensory maps (Lotze, 19th century). Skin patches learn their relative positions from the order in which they are stimulated during motion events. We propose that reversing the local motion within a global motion sequence ('motion scrambling') provides a good test for this idea, and present results of the first experiment implementing the paradigm. We used 6-point apparent motion along the forearm. In the Scrambled sequence, two middle locations were touched in reversed order (1-2-4-3-5-6, followed by 6-5-3-4-2-1, in a continuous loop). This created a double U-turn within an otherwise constant-velocity motion, as if skin patches 3 and 4 physically swapped locations. The control condition, Orderly, proceeded at constant velocity at inter-stimulus onset interval of 120 ms. The 26.4-minute conditioning (delivered in twenty-four 66-s bouts) was interspersed with testing of perceived motion direction between the two middle tactors presented on their own (sequence 3–4 or 4–3). Our twenty participants reported motion direction. Direction discrimination was degraded following exposure to Scrambled pattern and was 0.31 d' weaker than following Orderly conditioning (p = .007). Consistent with the proposed role of motion, this could be the beginning of re-learning of relative positions. An alternative explanation is that greater speed adaptation occurred in the Scrambled pattern, raising direction threshold. In future studies, longer conditioning should tease apart the two explanations: our re-mapping hypothesis predicts an overall reversal in perceived motion direction between critical locations (for either motion direction), whereas the speed adaptation alternative predicts chance-level performance at worst, without reversing.

## I. Introduction

*"When, in movement of the body, a stimulus changes its region of stimulation, the local signs change, and successive local signs are the things of adjacent localities"* (19th century philosopher Lotze, cited in [1], p. 268)

**Funding:** JLT Program grant APP1055084 National Health and Medical Research Council of Australia https://www.nhmrc.gov.au/ The funders had no role in study design, data collection and analysis, decision to publish, or preparation of the manuscript.

**Competing interests:** The authors have declared that no competing interests exist.

Somatosensory projection areas in the brain are dubbed brain maps because they reflect the topographical layout of the receptor surface. How maps develop and remain calibrated throughout life is a question of long-standing interest. It was empirically addressed in the classical study on synaptic plasticity by Merzenich and Jenkins [2]. In monkeys, they performed an anatomical (surgical) manipulation of the receptor surface by relocating a flap of skin from digit 4 to digit 3, fully preserving all its original innervations. Several months after the transfer, stimulation of the relocated flap excited cells in the cortical area that previously only represented the finger to which the flap was relocated. Since the surgery created new patterns of co-stimulation of different skin parts, the authors concluded that cortical representations are time-coincidence-based concepts.

In subsequent research, timing has usually been conceptualized and operationalized as temporal coincidence or neural co-activation (see [3–5]), although motion across the receptor surface is a better candidate for the general organizing principle of spatial maps (its importance was recently emphasized in [6–8]). It is a ubiquitous form of natural stimulation and, importantly, unlike simultaneous (coincident) stimulation, it cannot lead to 'fusion' of skin parts that often touch each other, such as lips or fingers. We therefore revive an old hypothesis (one source is quoted above) that *motion* organizes spatial maps in touch and vision. (There is of course a significant genetic component to topography, evident during embryonic development, but it is crude and insufficient [9]. There is also spontaneous, synchronized oscillatory electrical activity independent of interactions with the world, which occurs prenatally and in early development, when plasticity is high (in rodents and humans [9, 10]).

Motion is a very strong candidate for the experience-dependent map organization because locations next to each other on a sensory surface are stimulated one after the other by moving objects, and can thus learn that they are neighbours. The idea is illustrated in Fig 1 (and was previously described in [8, 9]). Although quite simple and old (Lotze, 19th C, cited in Herrnstein and Boring, 1965), it has attracted surprisingly little research and has little direct evidence to support it (to our knowledge, none in humans). Motion was used in two animal studies and it strongly influenced brain maps. One was a vision study, in which investigators reversed direction of optic flow in young tadpoles before the map was developed, resulting in a poor retinotopy. They concluded that *"visual information is transformed from a temporal code to a spatial code in the brain"* (p. 1, [11]). In another study [12], a rectangular flap of skin on a belly of the rat was rotated by 180 degrees, preserving the innervation as in the Merzenich study

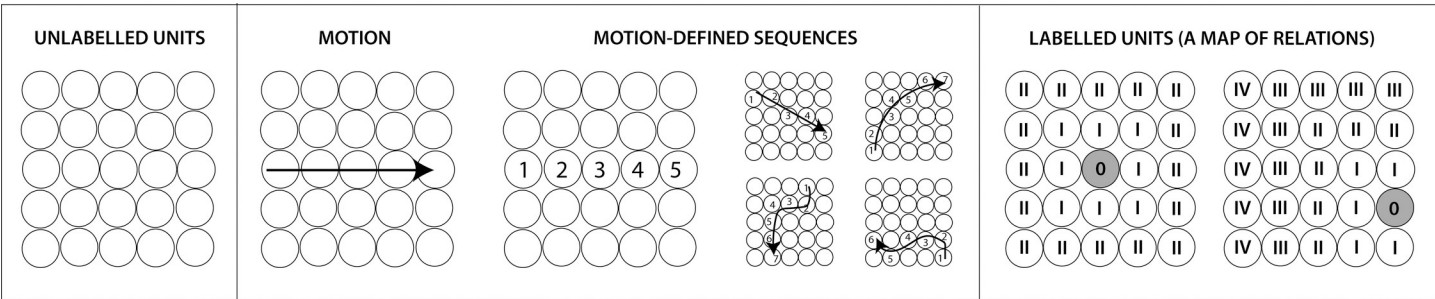

**Fig 1. An illustration of the idea that motion across a sensory surface informs about neighbourhood relationships.** *Left panel*: Units in a 2-D array (representing sensory neurons) have no 'labels' indicating their position in the array. *Middle panel*: Object motion proceeds in a sequence, activating units along its trajectory. Exposed to numerous motion events, adjacent localities will often be stimulated one after the other, as indicated by numerical sequences here. *Right panel*: As the outcome of this stimulation, units gain 'labels', i.e., the system gains information about their relative position in the array. Consider the central unit (labelled '0' in the picture on the left): its first-degree neighbours are the units stimulated immediately before or after (labelled 'I'); its second-degree neighbours ('II') are adjacent to its first-degree neighbours, etc. Each unit has a neighbourhood network (see a different unit depicted in the image on the right). Combination of these relationships makes a spatial map i.e., an array able to distinguish between different spatial configurations impinging on it.

described earlier. One group of animals was subsequently exposed to brushing stimulation across the line of skin incision for 7 hours, while control animals received no such input. The cortical neurons in the experimental group developed significant changes in receptive fields consistent with the new skin arrangement, unlike the control group, demonstrating that motion stimulation merely hours long is an effective stimulus for such a change.

The latter study is conceptually related to the present study, because we attempted to *simulate* skin re-arrangement. We have done so in conscious humans, using apparent (sampled) motion. Our aim is to provide support for the idea presented above, that any two locations are assigned their relative positions based on the order in which they are stimulated during object motion. By definition, the order of skin stimulation is consistent with motion trajectory: an object moving in a proximal direction–for example, up the forearm—will stimulate a more distal location before its proximal neighbor (Fig 2, left). However, by using discrete stimuli to produce apparent motion it is possible to reverse this order for two skin patches in the middle of the motion trajectory (continuous motion could be used to the same effect by reversing motion direction along a short segment within a longer trajectory). That is, during proximal motion, a more proximal patch can be stimulated before its distal neighbor (Fig 2, right, middle two locations), and vice versa.

If we did this for long enough, we would expect subsequent perceptual errors consistent with the swapping of places for affected locations. Continuing with the above example, if–following the conditioning—proximal location (P) was touched first and distal location (D) second, in a quick sequence, the *perceived* motion direction would be reversed (P←D, instead of P→D).

With shorter durations of conditioning, we would expect transitional effects wherein perception of whether motion was proximal or distal, and which location was proximal or distal, would be unclear.

The re-learning process would thus proceed through stages, from **(a)** *veridical* perception of motion direction (P→D) to **(b)** *degradation* of performance, which at its lowest point would be at chance level (P?D), and **(c)** *complete (illusory) reversal* of perceived direction (P←D). The equivalent changes with opposite sign would occur for real motion in the opposite direction. In a psychophysical study, stages **a-c** would be revealed through performance on a number of trials, where transition from **a** to **b** would appear as degraded performance (fewer correct responses), and stage **c** would manifest itself as an illusory reversal (majority of responses would be incorrect).

The aim of the present study was to determine whether the scrambled stimulus pattern results in predicted changes compared to a control condition, and if so, to what extent

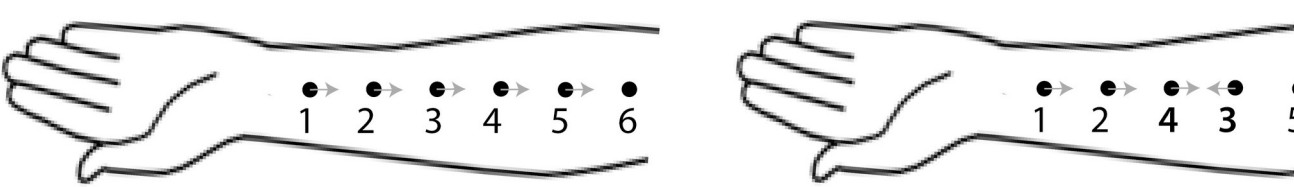

**Fig 2. Orderly and scrambled patterns of apparent motion across the skin.** Dots indicate touched locations. In both patterns, motion begins near the wrist and finishes near the elbow. They differ in the order of stimulation of the middle two skin patches only, as indicated by numbers. Grey arrows indicate direction of local stimulus motion i.e., motion between sequential stimulus pairs. Local motion has opposite direction to global motion in the middle of the motion trajectory in the Scrambled sequence.

perception changes due to our relatively short period of conditioning (approx. 26 min, interspersed with tests). We used d' as a measure of sensitivity to motion direction. Degraded performance (stage **b**) would manifest itself as decrease in d', and a complete reversal (stage **c**) would result in a negative d'.

We observed fewer correct reports about motion direction in the scrambled pattern compared to orderly, as expected, and consistent with transition to stage **b**. However, there was no complete motion reversal (d' remained positive in the scrambled condition), and that makes plausible an alternative explanation, as we explain in the Discussion. We also explain how longer conditioning times would help disambiguate the current findings.

## II. Method

We repeatedly applied a scrambled motion pattern or an orderly motion pattern to the forearms of human participants using a bidirectional conditioning stimulus (back-and-forth motion). Two-point motion between the middle two locations (see Fig 2) was the test, and participants reported the direction of motion.

### A. Participants

Twenty volunteers participated in the study (age range 18–30, 12 females), which was approved by the University of Sydney Ethics Committee. They were all naïve regarding study aims and design and were paid $20 per hour. All participants provided written consent prior to participation.

### B. Overview and study design

The experiment tested the ability to judge motion direction of a *test* stimulus following exposure to a *conditioning* stimulus. Study design of this repeated-measures experiment is described in Figs 3 and 4.

The participant's task was to report direction of apparent motion for stimuli applied by a pair of tactors. The direction was either proximal or distal. Perceived direction of the test stimulus obtained from this forced-choice task was the dependent variable. The main independent variable was the conditioning stimulation pattern (top row of Fig 3). In **Baseline**, these test stimuli were presented with no prior conditioning (Fig 4A, *Top left panel*). In the control condition, **Orderly** (Fig 2A; Fig 4A, *Middle panel*), the conditioning motion sequence proceeded

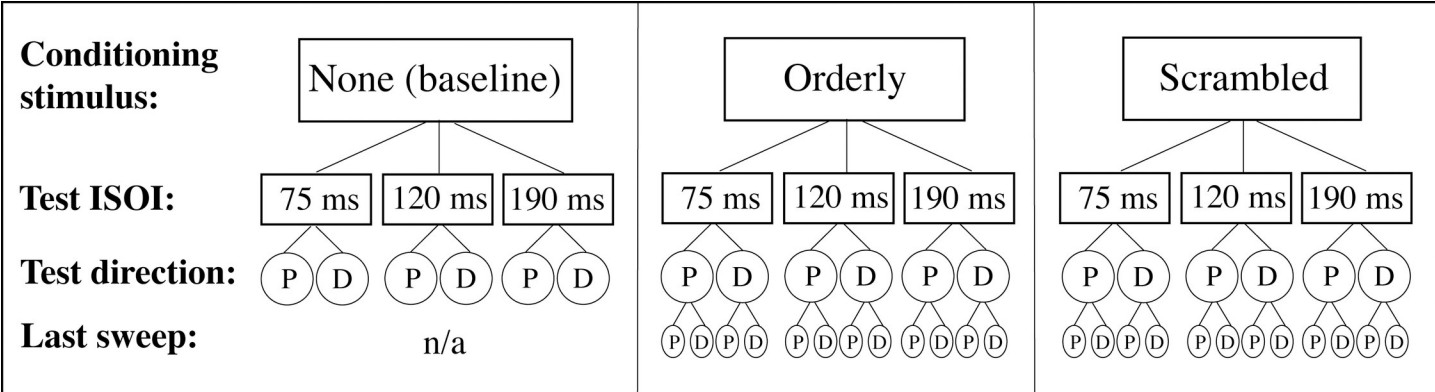

**Fig 3. Experiment design.** ISOI stands for Inter-Stimulus Onset Interval, P for proximal motion, and D for distal motion. 'Last sweep' refers to the last sweep in the conditioning sequence. See text for more details.

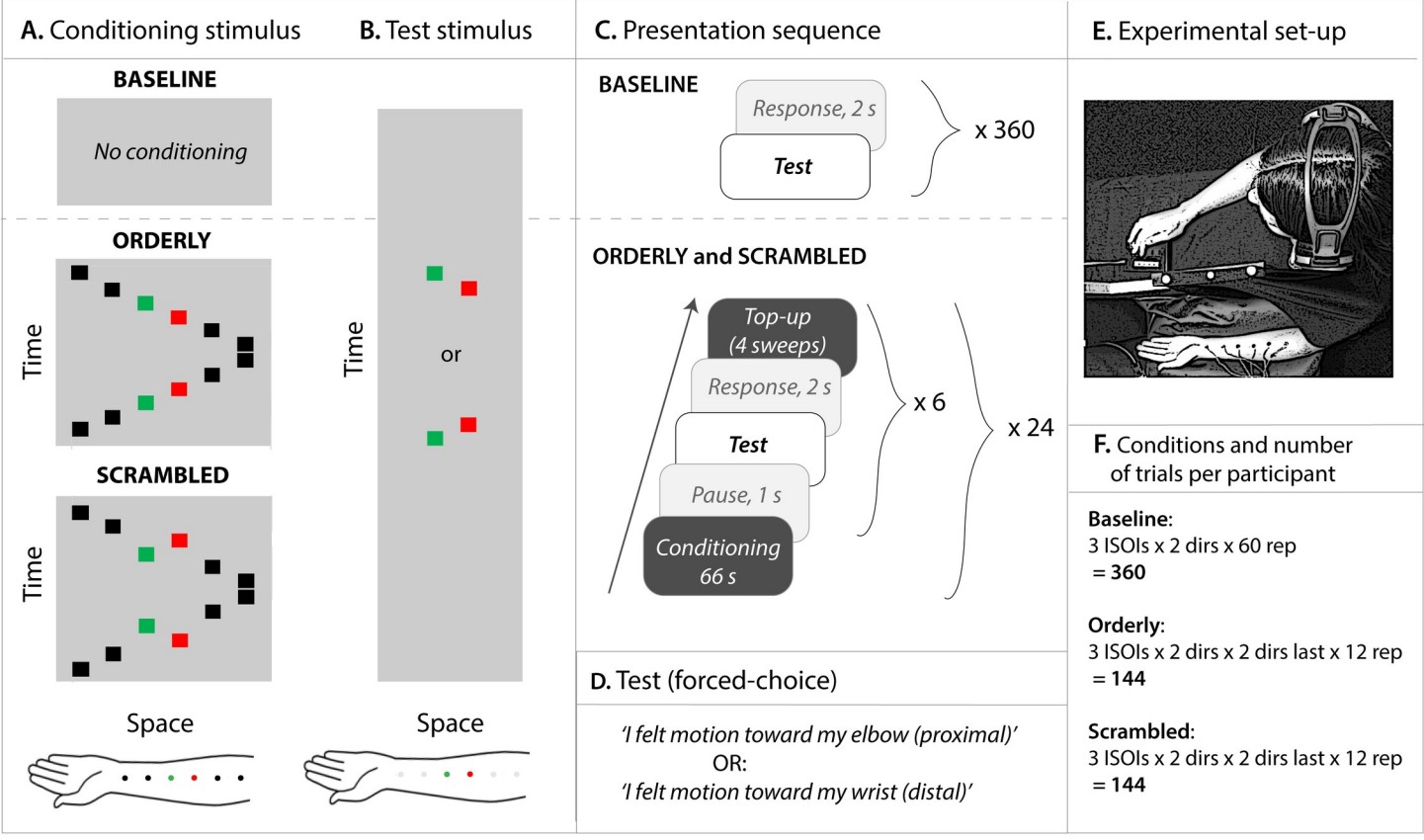

**Fig 4. Details of the method. A**. Conditioning was used in the Orderly and Scrambled conditions, shown here as space-time diagrams. Time in arbitrary units is represented on the Y-axis, and space (along the forearm) on the X-axis. Duration of one back-and-forth sweep was approximately 1440 ms. The black and coloured squares represent position of vibrators used in these conditions. **B**. Test stimulus, presented as time-space diagram, was the same in all conditions. The coloured squares represent vibrators used in these conditions and grey squares, vibrators attached to the forearm but not used. **C**. Stimulus sequence consisted of conditioning, 1-s break, test, 2-s for response, top-up, followed by five more repeats of the test-response-top-up cycle. **D**. Forced-choice task used to judge motion direction in the test stimulus. **E**. Bird's eye view of the experimental setup. Six vibrators were attached to participant's left forearm throughout the experiment, occluded from participant's view. White noise presented through headphones masked the sound of vibrators. The participant responded to test stimuli by pressing one of the two buttons on the response box. **F**. Conditions and number of trials per participant ('dir' stands for direction of the test stimulus, 'dir last' for the direction of the last conditioning sweep, and 'rep' for the number of repeats per condition). Vibrator placement, our second control variable, is not explicitly presented for simplicity; it is embedded in repeats, such that each of the vibrator orders was used in half of the repeats.

across the skin at constant velocity (1-2-3-4-5-6, or 6-5-4-3-2-1). In the **Scrambled** condition (Fig 2B; Fig 4A, *Bottom panel*), the activation sequence for locations 3 and 4 was reversed (1-2-**4**-**3**-5-6-5-**3**-**4**-2-1). The inter-stimulus onset interval (ISOI) in the conditioning sequence was always 120 ms, and the inter-stimulus interval (ISI) was 0 ms.

Following conditioning with either pattern, test stimuli comprising locations 3 and 4 were presented in the order 3–4 or 4–3. The second independent variable was Inter-Stimulus Onset Interval (ISOI) for the test stimulus (second row of Fig 3). Three values chosen after extensive piloting were **75, 120** and **190 ms,** while piloting itself was informed by the literature (see [13, 14]). Our intention was to include stimuli difficult to discriminate—as was the case for the ISOI of 75 ms, and those relatively easily to discriminate (190 ms ISOI). Duration of vibration at each skin location was equal to the ISOI. For example, in the 75 ms condition, each vibration lasted 75 ms, immediately followed by vibration at the next location. Pilot studies suggested that zero-ISI results in the smoothest perception of motion. The third independent variable was direction of motion in the test stimulus: **proximal** or **distal** (third row of Fig 3; Fig 4B).

We controlled for the direction of the last sweep in the conditioning stimulation (fourth row of Fig 3). Motion direction in the conditioning stimulation alternated: each sweep in one direction was followed by the sweep in the opposite direction. This should have created equal net adaptation and no net directional aftereffect (directional aftereffect in tactile motion is the bias to perceive direction opposite to the preceding motion; see [15, 16]), except for the possible greater influence of the last sweep in the conditioning stimulus, which was followed by the test. To control for the direction of the last sweep, half of conditioning trials ended with a proximal sweep, randomly interleaved with distal last sweeps.

The final variable we manipulated (the second control variable) was placement of the vibrator array (not shown in Fig 3). Vibrators were numbered 1–6, and in one half of each participant's sessions, they were physically placed in order 1–6 proximo-distally (from near the elbow crease toward the wrist), and in the other half, in reversed order (6–1). Which order was used in which half-session was counterbalanced across subjects.

The total number of different experimental conditions was 30: 6 in Baseline (3 ISOIs x 2 directions of test motion) and 12 each in Orderly and Scrambled (3 ISOIs x 2 directions of test motion x 2 directions of the last conditioning sweep). Perception of motion direction in the test stimulus was assessed using a forced-choice task.

## C. Apparatus, set up and procedure

Baseline, Orderly and Scrambled stimuli were presented in separate sessions, on separate days, in a partially counterbalanced order across participants (counterbalancing was imperfect because we had 20 participants and could thus not have an equal number of all possible orders of the three conditions). Presentation sequences are illustrated in Fig 4C. No conditioning stimulus was presented in the Baseline condition, and test stimuli were presented one after another, separated by a two second break for response, divided into two equal blocks of 180 stimuli with a break in-between. In the Orderly and Scrambled conditions, participants initially received 66 seconds of conditioning with stimulus motion up and down the forearm. After a one-second break, they were presented with one test stimulus and had two seconds to report direction of apparent motion in the test stimulus ('proximal' or 'distal', see Fig 4D) using a response box. Immediately afterward, they received top-up conditioning consisting of two sweeps up and down the forearm, followed by another test. Six test stimuli were presented in this manner, separated by top-ups. After a short break, the whole cycle was repeated (66 s of conditioning, and six test-response-top-up sequences). This was repeated 24 times in the Orderly and Scrambled sessions, divided into two blocks of 12 each, with a 10-minute break in between. Placement of the vibrator array (the first control variable described earlier) was different in the two blocks.

The radial aspect of the forearm, hidden from participant's view (see Fig 4E), had a linear array of 6 coin-motor vibrators attached to it, placed 4 cm apart, centre-to-centre. Activating vibrators one after another created perception of apparent motion. The ISOI in the conditioning stimuli was 120 ms, equal to duration of vibration. One sweep up or down the forearm lasted 720 ms. The test stimulus was presented in the order 3–4 or 4–3, at one of the following ISOI: 75 ms, 120 ms and 190 ms, (corresponding to the velocities of 53.3, 33.3 and 21.1 cm per second, respectively).

The vibrators used to create a sense of apparent motion were 10 mm in diameter, 3 mm high cylindrical coin motors (Precision Microdrive™), in which eccentric rotating mass results in vibration. They were controlled by a custom developed software (LabView™ 2012). We used a laser apparatus (OptocoNCDT 2200; data extracted using LabChart™) to measure vibration frequency and the degree of vibration transmission to the surrounding skin. Vibration frequency was initially unequal for different vibrators, but adjustment of current brought

them all to approximately 110 Hz (to control for any remaining differences, the order of vibrators was reversed in half of the trials, as described earlier). Laser measurements also showed that vibration transmission via the skin occurs over at least 4 cm distance from the vibrator. This was consistent with a perception test, in which a fingertip was placed at different distances from the vibrator attached to another person's forearm. Skin vibration was in some instances detectable 8 cm away from the vibrator, double the 4-cm separation between the vibrators we used. Thus, the stimulus delivered to a particular location affected a much greater area, adding noise to our desired spatiotemporal stimulus pattern (but we don't know how much of the vibration spread was above threshold).

As Fig 4F shows, the total number of trials in Baseline was 360 (3 ISOIs x 2 directions x 60 repeats), and in Orderly or Scrambled, 144 (3 ISOIs x 2 directions x 2 directions of the last sweep x 12 repeats). Each participant thus completed a total of 648 trials.

Each session was preceded by a short practice, which differed between conditions. In the practice for the Baseline condition, 90 test stimuli were divided into three blocks separated by two short breaks. Practices for Scrambled and Orderly conditions began with 60 test stimuli, followed by three bouts of conditioning containing 6 tests each.

A short questionnaire was used at the end of Scrambling and Orderly sessions to capture participants' perception of the 66-s conditioning stimulus. The participants ranked the frequency of experience on a 7-point scale ranging from 'never' to 'always'. Example statements are: *During the longer (1-min) periods of stimulation, I felt. . . 'Motion on my forearm'; 'Motion along the straight line'; 'Curved or zig-zag motion path'.* The order of questions was randomized for each participant.

Participants also answered two open-ended questions and gave a phenomenological report by sketching what they had felt on a standardized picture of an arm.

## D. Data analysis

Raw data were responses regarding the direction of motion (distal vs proximal) of the test stimuli. R (R Foundation, R 3.1.0, 2014) was used to extract data and compute the proportion of correct responses for each participant in each condition, and SPSS for further analysis. There were 12,960 possible responses (20 participants x 648 responses), of which 217 or 1.7% were missing (failed to record–either the participant did not respond, or they pressed the response button with insufficient force).

We used signal detection theory to compute sensitivity to motion direction (d') and bias (c). This absorbed one of our independent variables, direction of test motion. D-prime was defined as the difference between z-scores for the proportion of correct responses to proximal test motion and the proportion incorrect to distal test motion. Bias was the average of the same pair of z scores, multiplied by -1 ('c' measures of bias [17], p. 143). Computed in this manner, negative c means bias toward the response 'proximal'.

The example in Table 1 shows steps in computing d' and c for one experimental condition (Scrambled, 120 ms ISOI) in one participant. Note that the four values each of d' and c (shown in the last two rows of Table 1) were computed based on a total of 48 stimuli presented, 6 repeats for each of the 8 conditions specified in the table (2 directions of test motion x 2 Vibrator placement options x 2 Directions of the last conditioning sweep). All four values were used in linear mixed modelling (we did not average them) to allow us to test for quadratic trends across the three ISOIs. Variability between the four measures of d' and c included variations, if any, caused by our two control variables (Vibrator placement and Direction of the last sweep).

A further 30 responses were excluded (single participant data, Baseline, 120 ISOI, Vibrator placement 1–6) because d' and c computed from them were both outliers (beyond 2.5 standard

**Table 1. Example computation of d' and c, scrambled condition, 120-ms ISOI, one participant.**

| | Test stimulus motion (6 stimuli per condition) | | | | | | | |
|---|---|---|---|---|---|---|---|---|
| | Proximal stimulus motion | | | | Distal stimulus motion | | | |
| Placement of vibrators 1 to 6 | Elbow to wrist | | Wrist to elbow | | Elbow to wrist | | Wrist to elbow | |
| Last conditioning sweep | dist. | prox. | dist. | prox. | dist. | prox. | dist. | prox. |
| Correct response | 6 | 4 | 5 | 3* | Redundant (total responses per condition = 6) | | | |
| Incorrect response | Redundant (total responses per condition = 6) | | | | 4 | 3 | 1 | 0 |
| Hit rate for Proximal motion | 1.00 | 0.67 | 0.83 | 0.60 | | | | |
| False Alarm (FA) rate for Proximal motion | | | | | 0.67 | 0.50 | 0.17 | 0.00 |
| Hit rate for Proximal motion, corrected (0 becomes 0.05; 1 becomes 0.95) | 0.95 | 0.67 | 0.83 | 0.60 | | | | |
| FA rate for Proximal motion, corrected (0 becomes 0.05; 1 becomes 0.95) | | | | | 0.67 | 0.50 | 0.17 | 0.05 |
| Z score for corrected Hit rate | 1.64 | 0.43 | 0.97 | 0.25 | | | | |
| Z score for corrected FA rate | | | | | 0.43 | 0.00 | -0.97 | -1.64 |
| Ability to discriminate proximal from distal test motion (d' = $z_H - z_{FA}$) | 1.21 | 0.43 | 1.93 | 1.90 | | | | |
| Bias toward response 'distal' (c = $-(z_H + z_{FA})/2$) | -1.04 | -0.22 | 0.00 | 0.70 | | | | |

Note that test motion had two directions, proximal (toward the elbow) and distal (away from the elbow). Placement of vibrators 1–6 and Last conditioning sweep are control variables described in the text (Section B, Methods).

*One response was missing in this condition; thus 3/5 = 0.60 hit rate.

deviations from their respective means). The corresponding Scrambled and Orderly conditions were also excluded because they could not be corrected for baseline in the further analysis.

To account for change in direction discrimination due to adaptation from the conditioning, we compared Baseline with the other two conditions. Our main question was addressed by the sensitivity analysis: we expected conditioning with Scrambled motion to result in reduced sensitivity (more motion reversals) compared to Orderly. For this analysis, we subtracted d' in Baseline from each of the other two conditions, and compared Orderly and Scrambled to each other only. We analyzed response bias in a similar manner.

Linear Mixed Modelling (LMM) for repeated measures data [18] was performed via GLM procedure in SPSS (v. 24). LMM accounts for the repeated nature of the data and for random variation across individuals. It also allowed the independent variable ISOI in the test stimulus to be treated as a continuous measure. Fixed factors were Conditioning pattern, ISOI (and its quadratic term) and their interaction. Participants were treated as a random factor, removing a significant proportion of within-subject covariance from the residuals; the Repeated subcommand in LMM dealt with the remaining deviations from the assumptions of a linear model.

Our approach to modelling is to begin with the full model, including independent variables and multiplicative terms of interest (quadratic trends, interactions), and random and repeated statements. We first adjust the random and repeated statements, and then the fixed factors. The model choice was guided by the AIC criterion and parsimony. It had to include the variables and interactions of central interest but not all possible interactions (e.g., we tested but excluded the interaction between Conditioning pattern and the quadratic term for ISOI because it did not improve the model and was not of particular interest). Both d' and c were analyzed in this manner. Data files and SPSS syntax files that include all models we tested are

available at the Open Science Framework data repository. Interested reader can find more details about model development in the comments within the syntax files.

Questionnaire results were expressed on 0 ('Never') to 6 ('Always') scale, and summary measures were compared across conditions.

## III. Results

### A. Sensitivity to motion direction

Detailed results for Baseline, Orderly and Scrambling conditions are shown as box plots in Fig 5.

The ability to discriminate direction was highest in Baseline (white boxes in Fig 5, *Left panel*). It increased with ISOI, approaching the ceiling at 190 ms. Sensitivity was lower both in Orderly (light grey) and Scrambled (dark grey) compared to Baseline.

Our critical result is shown in Fig 5, *Right panel*: reversed direction of the test stimulus (opposite to that actually presented) was perceived more frequently in Scrambled condition compared to Orderly. This is indicated by lower sensitivity (lower d') in that condition. Note that d' values in Fig 5, Right are negative because they are shown relative to Baseline–they simply show that performance was worse than in the Baseline (they do not indicate that perceived direction was reversed overall).

Linear mixed modelling (LMM) was used to estimate the difference in sensitivity (d') between Orderly and Scrambled conditions after each of them was corrected for Baseline. Estimated quadratic functions are shown in Fig 5, *Right panel*, and regression coefficients are given in Table 2. The effect of Conditioning pattern was significant ($F(1, 78.2) = 7.82$, $p = .007$), as was the ISOI ($F(1, 49.5) = 10.68$, $p = .002$). Sensitivity for motion direction was lower

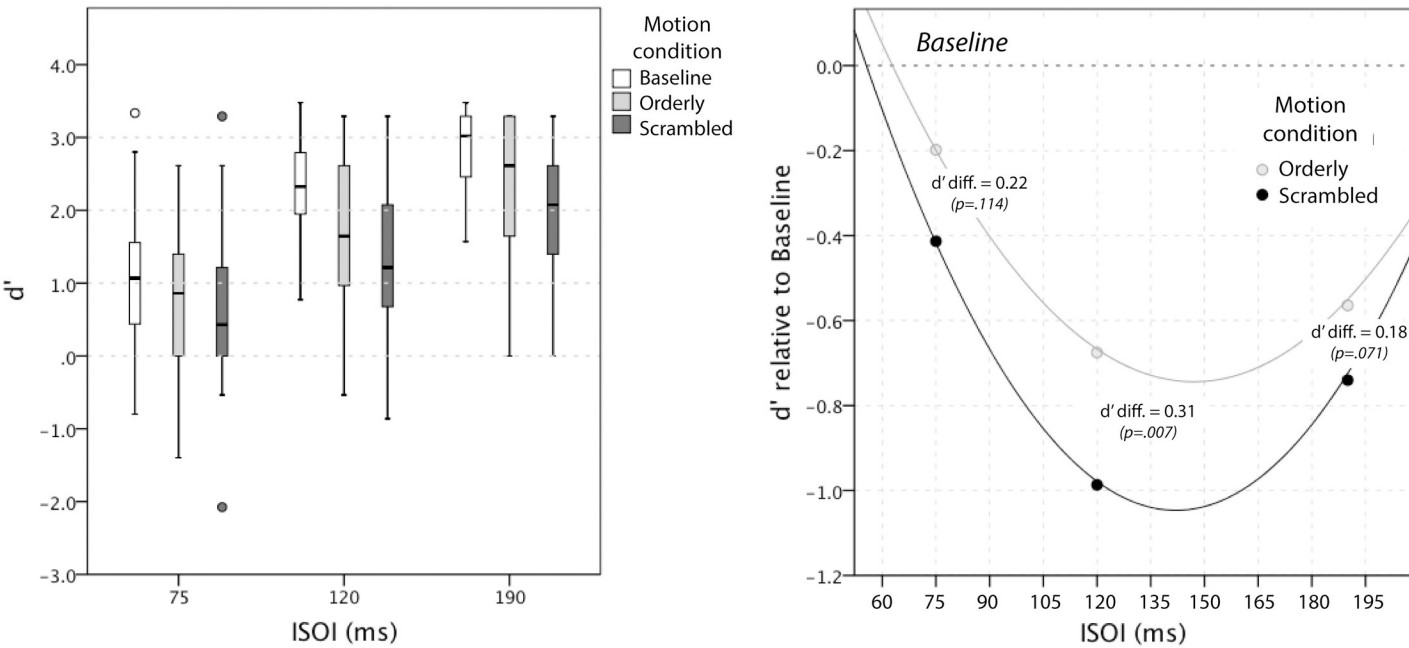

**Fig 5. Sensitivity to motion direction, results.** *Left panel.* Box plots show medians and variability in d' for 20 participants as a function of ISOI and Motion condition. Note (a) the advantage of Baseline at all ISOIs, (b) the advantage of Orderly over Scrambled at all ISOIs, and (c) a ceiling effect at 190 ms, most pronounced for Baseline. *Right panel.* Estimated marginal means from LMM analysis of Baseline-corrected results (d-prime values are negative because Baseline was superior to both Orderly and Scrambled). The fact that Scrambled stimulation produced higher negative d' values indicates worse discrimination of direction compared to Orderly. See text for details.

**Table 2. Results of linear mixed modelling (LMM), analyses of d' and bias (c); the critical result is the effect of motion condition on d' (orderly vs scrambled).**

| Analysis of sensitivity (d') | | Analysis of bias (c) | |
|---|---|---|---|
| | **Dependent variable: d'** | | **Dependent variable: c** |
| *Fixed factors* | *Coefficients (95% CI)* | *Fixed factors* | *Coefficients (95% CI)* |
| Intercept | -0.9786 (-1.2312; -.73) | Intercept | 0.1076 (.0292; .1861) |
| ISOI (centered; 120 = 0) | -0.0062 (-.01; -.002) | ISOI (centered) | 0.00082 (-.00022; .00187) |
| Scrambled | reference | Scrambled | reference |
| Orderly | 0.3114 (.0897; .5332) | Orderly | -0.01272 (-.06593; .04049) |
| Orderly x ISOI | 0.0005 (-.0036; .0043) | Orderly x ISOI | *Tested and excluded from the model* |
| Scrambled x ISOI | reference | Scrambled x ISOI | |
| ISOI$^2$ | 0.0001 (.00006; .00021) | ISOI$^2$ | |
| Orderly x ISOI$^2$ | 0.000035 (-.00011; .00004) | Orderly x ISOI$^2$ | |
| Scrambled x ISOI$^2$ | reference | Scrambled x ISOI$^2$ | |

See text for details.

in Scrambled than Orderly condition. Compared to Baseline, it changed with ISOI following a quadratic trend (F(1, 77.3) = 15.15, p < .001), and the change was greatest for the ISOI of 120 ms. The interaction between Motion condition and ISOI was not significant (F(1, 76.5) = 0.06, p = .801), nor was it its interaction with ISOI$^2$ (F(1, 75.9) = 0.86, p = .357).

Estimated differences (and associated 95% CIs) between d' in Orderly and Scrambled from this model were 0.22 (-.05 to .49), 0.31 (0.09 to 0.53), and 0.18 (-.02 to .37) at 75 ms, 120 ms, and 190 ms, respectively, with the following p values for Bonferroni-corrected pairwise comparisons: .114, .007 and .071.

## B. Bias in judgments of motion direction

Detailed results for the three motion conditions are shown in Fig 6. Summary of raw values of c is shown on the left, and absolute values on the right. Absolute values were computed because

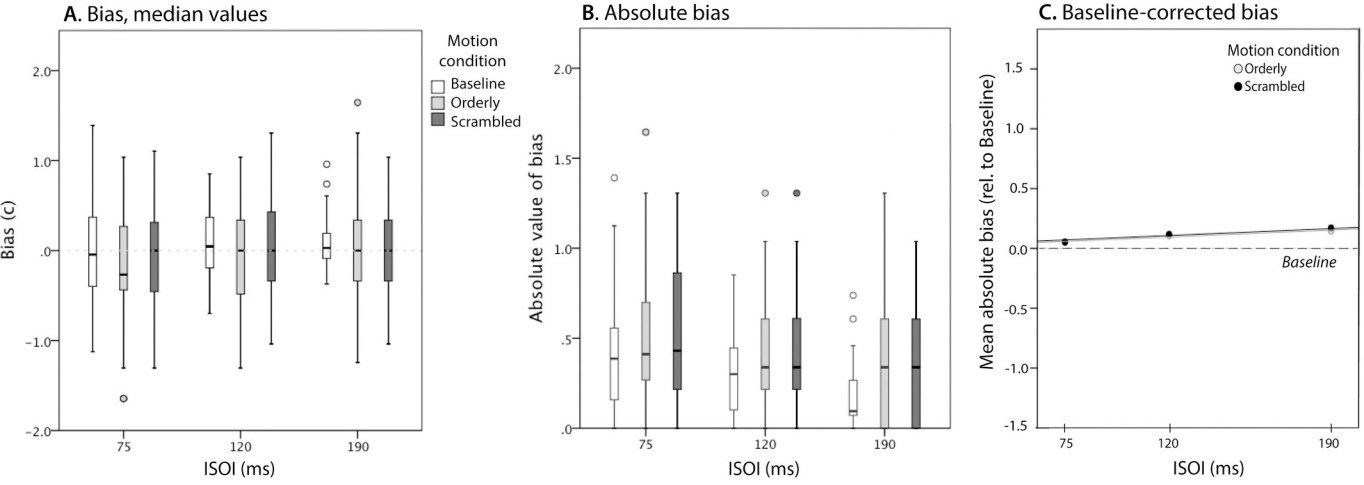

**Fig 6. Bias in judgments of motion direction. A**. Bias as a function of ISOI and Motion condition. Positive bias is tendency to report distal motion. Note that most medians are close to zero. **B**. Absolute values of bias, computed separately for each participant and condition. **C**. Baseline-corrected absolute bias for Orderly and Scrambled conditions, group means and linear functions estimated using linear mixed modelling. Note that the lines almost completely overlap, and that both Orderly and Scrambled conditions produced slightly more biased responses than Baseline.

proximal and distal bias cancel each other out, potentially misrepresenting the strength of each individual's bias and its variation across conditions.

There was almost no systematic bias at the group level (Fig 6A), with little variation across conditions. Absolute values (Fig 6B) show more bias and more variation. The condition with least absolute bias was Baseline at 190 ms (the stimulus easiest to judge–see Fig 5, *Left panel*).

Baseline-corrected Orderly and Scrambled absolute bias is shown in Fig 6C. Slightly greater than in Baseline (represented by a dotted line), bias is very similar in the two conditions: their means and linear functions estimated using LMM practically overlap. The effect of Motion condition was not statistically significant (F(1, 79.5) = 0.226, p = .635). A mild increase in bias with ISOI estimated by the model (0.082 per 100 ms) was also not statistically significant (F(1, 18.2) = 2.726, p = .116).

## C. Phenomenological reports

Answers to the questionnaire designed to explore perception of the conditioning stimuli are summarized in Fig 7. It shows medians and standard errors for 19 participants (one participant's data are missing due to experimenter error). Orderly and Scrambled conditioning stimuli were experienced similarly: all participants in both conditions felt motion up and down the forearm, mostly along the straight line, with occasional irregularities in the motion path (gaps, curves, zig-zag motion, twists and turns). Most of the time, it appeared to them that one object was moving, and sometimes two or more.

A one-point median difference was found for questions 5 ('I felt motion along the straight line') and 9 ('I felt as if a single object was moving'), both more frequently experienced in the Orderly condition. Half a point median difference was found for question 3: 'I felt a discontinuous motion path (with gaps)', more frequently experienced in the Scrambled condition.

Drawings were scrutinized for any systematic differences between the two conditions, including presence of gaps and other irregularities, but there was no clear trend. All 20 pairs can be seen in the Open Science Framework data repository [https://osf.io/gtcr7/?view_only=87c6fb7b513b49758aa7185dcdf0e984].

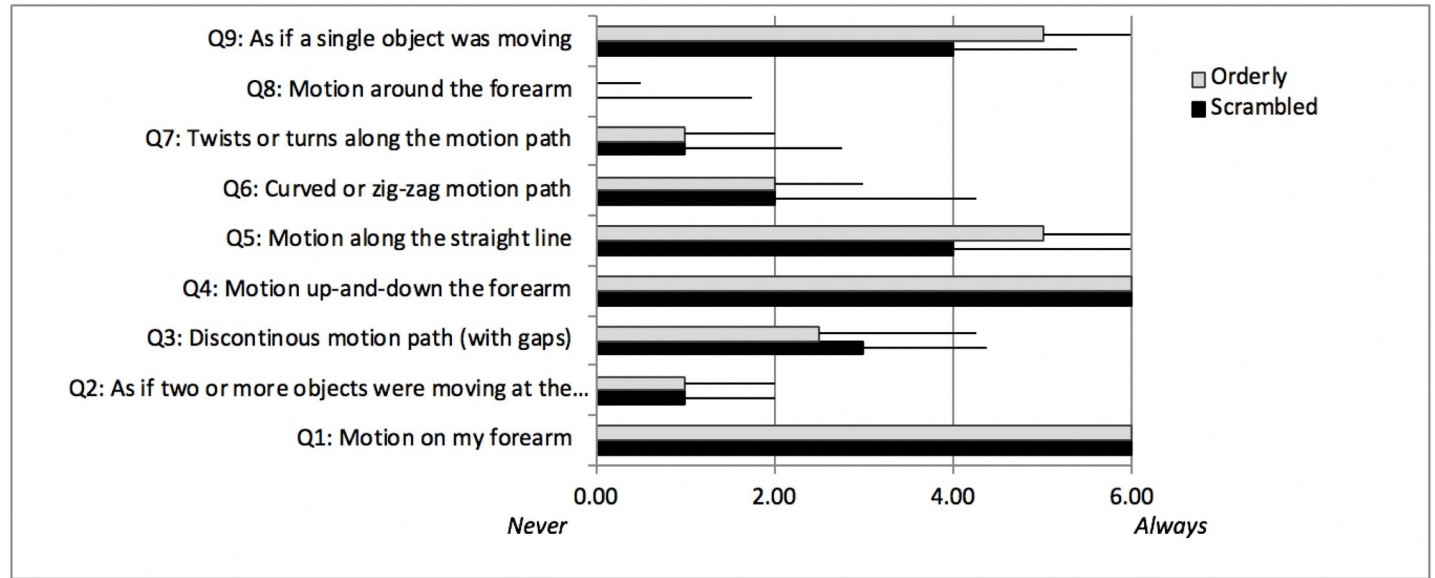

**Fig 7. Answers to the questionnaire designed to explore perception of the conditioning stimuli (medians and standard errors, n = 19).** To ensure participants were referring to the conditioning stimulus rather than test stimulus, the root question asked: '*During the longer (1-min) period of stimulation, I felt. . .*').

## IV. Discussion

Sensory systems respond to their 'diet' (see [19] and [20] for an early and a recent reference). Sufficient exposure to a new diet should result in a change, provided the system responds to the altered aspects of the diet. The sensory diet we provided was bi-directional apparent motion lasting 26.4 minutes in total (per session), created using discrete vibration stimuli and delivered in 66-s bouts of conditioning, interspersed with tests in either direction. In response, both scrambled and orderly conditioning was followed by a reduced ability to discriminate motion direction in the test stimulus relative to baseline. It was reduced at all test speeds, and mostly so for the 33 cm/s test, which matched the adapting speed (ISOI = 120 ms, see Fig 5 *Right*). The crucial aspect of the diet we varied was the order of stimulation of the middle two skin patches, which were also the test patches. The scrambled sequence resulted in worse test performance than the orderly sequence, as predicted. The difference was again greatest for the 33 cm/s test.

Qualitative data (see Fig 7) and the drawings show that the two conditioning patterns were similarly perceived: the participants felt motion up and down the forearm, mostly along a straight line. A variety of tactile and visual spatiotemporal patterns containing sudden accelerations are misperceived such that the percept tends to be smoother than the stimulus patterns [6, 7, 21, 22]. Vision research shows that transient changes in the motion sequence such as gaps in the trajectory or changes in colour or shape of the moving object are imperceptible provided they do not occur too early in the motion sequence [23].

Why were different results obtained in the orderly and scrambled conditions?

### Possible causes of different adapted states in the orderly and scrambled conditions

Intensity adaptation [24] and adaptation to motion [15, 16, 25, 26] can potentially both account for the impaired ability to distinguish proximal from distal motion in both the orderly and scrambled conditions compared to baseline. However, our main prediction was that conditioning with the scrambled pattern would cause an *even worse* performance in a subsequent test than the orderly pattern. This prediction was confirmed. We stimulated exactly the same skin locations an equal number of times in both conditions, therefore the intensity adaptation alone cannot explain the difference. The explanation must lie in the temporal sequence of stimulation (i.e., motion pattern). In the introduction, we argued that sequences of stimulation caused by object motion across the skin define relative positions of elements within a somatosensory map (illustrated in Fig 1). In what follows we delve further into that explanation, followed by other potential explanations of the present results (not mutually exclusive).

**1) Adaptation as the beginning of map change due to a new diet of motion patterns.** The idea we explore is that elements in a map get assigned their relative positions based on the order in which a moving object stimulates them. We reversed motion direction over locations 3 and 4, creating a local motion opposite in direction to the global motion, as if the order of skin patches underneath vibrators 3 and 4 were actually swapped. The results are consistent with an adaptive process that began to *re-assign relative positions* of somatosensory neurons with receptive fields in locations 3 and 4 accordingly. Our findings do not show full reversal (stage **c** described in the Introduction) but could represent the beginning of the process (stage **b**).

Is the proposed process feasible? Given that objects in the world often accelerate, if accelerations were to cause map change, a consequence could be instability in neural networks. However, this should not happen with the proposed mechanism, because the acceleration needs to *consistently occur on the same segment of the sensory surface*. It is crucial for the process we

propose that acceleration and skin location are thus correlated. Map reorganization results in decorrelation and a better match with the world.

The principle of decorrelation is one widely considered principle of efficient sensory coding. It is proposed to be a rectifying (or preventative) self-organizing process in neural populations [27–29]. The correlation–and decorrelation—of interest to us here is not between features of the external stimulus, but between *stimulus acceleration and its location* on the sensory surface. A similar idea was put forward by New and Scholl [30] to account for motion-induced blindness, a phenomenon where a small object always falling on the same segment of the retina amidst a dynamic visual field quickly fades from awareness. They interpreted it as "the visual system's attempt to separate distal stimuli from artifacts of damage to the visual system itself" (p. 655).

Neural mechanisms supporting the proposed changes likely involve context-sensitive, long-range connections between neurons in sensory maps and feedback from higher-order motion neurons on neurons that encode local motion and position. Involvement of the long-range connections allows the context of stimulation to disambiguate local input [31]. Filling-in of blind spots in vision and deafferented skin areas (numb spots) relies on such connections [32, 33] and blind spots are conceptually similar to our scrambled stimulus: both create discontinuities in the sensory input and both are 'glued' to a certain position on the receptor surface. Changes were observed in receptive fields in cats under comparable stimulation regimes within minutes of stimulation, resulting in receptive field increase by the average factor of 5.2 [33]. Rapid changes were also shown in S1 activity patterns following minutes of correlated finger stimulation in humans [34], suggesting dynamic receptive fields that adjust to specific stimulation patterns.

The term 'adaptation' is typically used for relatively short-lived changes, and 'plasticity', for more profound structural map changes that we propose would eventuate. It is not always easy to draw a line between the two types of processes—they may overlap (see [35], Box 1). We propose they would overlap in the present case: should stimulus conditions persist, the present rapid adaptation would gradually lead to a profound change, reflected in altered performance across a number of spatial tasks. While we emphasize spatial rather than temporal tasks as tests of a possible change, temporal order judgment may also change as the result of exposure to Scrambled motion sequence.

Other types of adaptation may possibly also account for the effect we observed, and we turn to them now.

**2) Adaptation as reduced responsiveness (gain reduction) due to exposure to motion.** Both scrambled and orderly patterns gave rise to perception of motion (see Fig 7). Their repeated presentation would have activated motion-sensitive neurons, which have a strong presence in S1 [36, 37] and many of which are sensitive both to direction and speed [38–40]. Our conditioning was bidirectional so both directions should have been similarly adapted, resulting in no net motion aftereffect.

The question of interest here is whether velocity adaptation in scrambled pattern could have been the cause of *worse* performance in a subsequent test than the orderly pattern. Adaptation effects are complex and occur at multiple levels (see [20] for review). Neurons with receptive fields within the test area (affected by vibrators 3 and 4) would have adapted, and so would those with larger receptive fields, extending into the surround area. We know that faster moving tactile stimuli create stronger adaptation, reducing subsequently perceived speed [25] and increasing directional aftereffect [26]. This suggests two possible adaptation effects, one predicting *worse performance in the scrambled* pattern (consistent with the results). and the other predicting *worse performance in the orderly* pattern (opposite to our results).

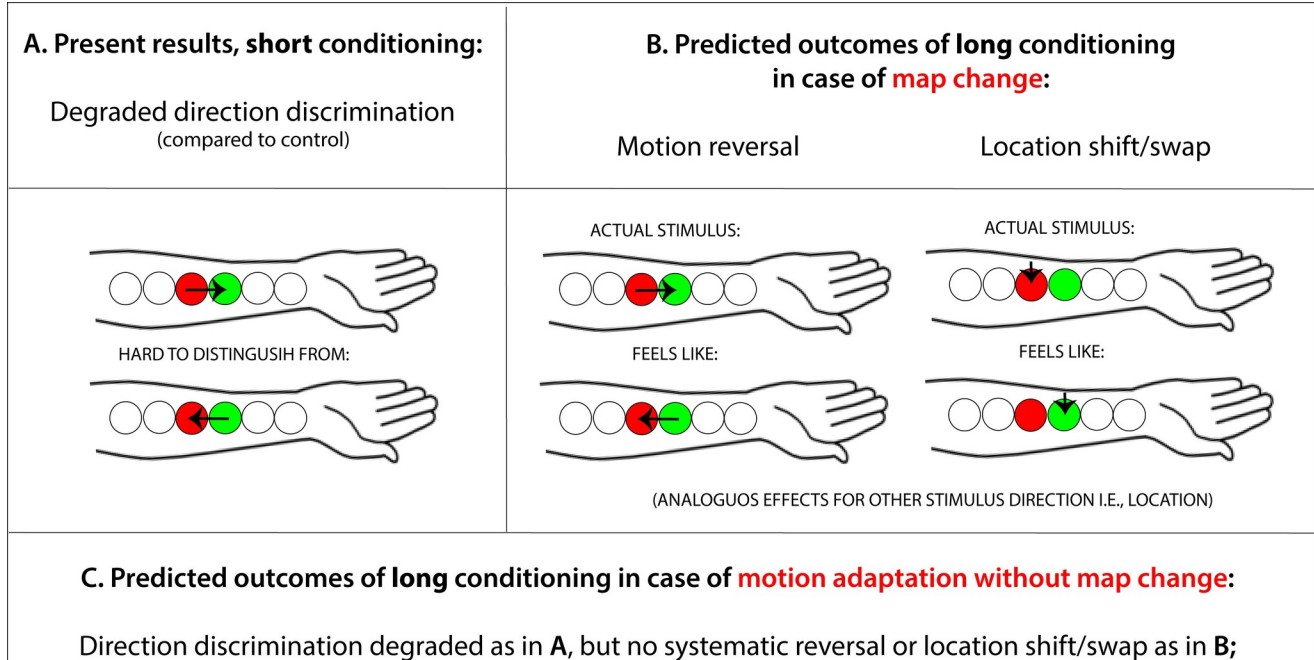

**Fig 8.** Expected outcomes of long exposures to scrambled motion differ in case (A) of speed adaptation alone, which would simply deepen motion confusion found in the present study, and (B) of map change, in which case we expect motion reversals and locations swap.

First, *higher average speed* in the scrambled condition might have resulted in *stronger* adaptation in that condition. Motion between the test locations 3 and 4 was of equal speed in scrambled and orderly patterns, but speed between locations 2 and 4, and 3 and 5, was two times greater in the scrambled pattern. This may have increased the adaptation level in motion neurons with receptive fields larger than the span from locations 3 to 4. Faster adapting speed may thus have made the direction of the subsequent stimuli harder to detect.

Second, *frequent direction change* in the scrambled condition might have resulted in *weaker* motion adaptation in that condition. Unlike the orderly pattern, in which direction of motion was constant during a single sweep–from elbow to wrist, or vice versa—in the scrambled pattern, direction change occurred twice during each sweep, in a double-u-turn. Perhaps direction-sensitive neurons whose receptive fields cover the whole motion trajectory adapted less when direction thus changed in the scrambled condition. In vision, global motion neurons in area MT integrate inputs from local motion detectors [41], and respond more strongly to more coherent motion in their preferred direction [42]. Since the Scrambled pattern presents *less* coherent motion within the same time period than Orderly, it is possible that it results in less adaptation. This is a conjecture; a detailed analysis of motion integration is not available for tactile motion-sensitive neurons with large receptive fields on the forearm (neurons themselves were described in [36]). However, there are similarities between areas MT in vision and Brodmann's area 1 in touch [43] and it is possible also that global motion neurons in touch are similar to those in vision.

Stimuli such as ours have not been used before and the presence and extent of these two proposed adaptations cannot be predicted from prior research or estimated from the present results. If they occurred, they might have cancelled out each other, or one might have prevailed.

Longer conditioning is required to see how the changes we observed would evolve, which in turn would allow their clearer interpretation (illustrated in Fig 8). Fig 8A shows degraded

direction discrimination we obtained following short conditioning. If our hypothesis is correct, very long motion conditioning (extending to days and weeks) would simulate a surgical swap of locations 3 and 4 (as in [2] and [12]), eventuating in perceptual changes: motion direction (or temporal order) judgment should be completely reversed, and there should also be corresponding errors in absolute localization (illustrated in Fig 8B). On the other hand, if instead of motion reversals and other predicted effects, long-term conditioning only led to further degradation in motion discrimination, that would weaken our hypothesis and strengthen the speed-adaptation explanation (see Fig 8C).

## V. Conclusion

Our motion-scrambling paradigm is a novel and potentially useful tool in the psychophysical study of adaptation and plasticity in sensory maps in conscious humans. It simulates re-arrangement of skin patches. We propose that prolonged exposure to such re-arrangement would change perceived spatial relationships of the skin segments along the motion path, triggered by accelerations tied to a particular location on the sensory surface.

We observed degraded direction discrimination between our two scrambled locations. This is consistent with the above proposal, but inconclusive. It could mark a beginning of the map change, but it could also be due to speed adaptation in direction-sensitive neurons. Longer conditioning using the scrambling stimulus would allow us to distinguish relative contributions of different adaptation processes.

## Acknowledgments

We thank Raymond Patton for constructing the vibrator array, and Timothy Turner and Diego Barneche for custom software.

## Author Contributions

**Conceptualization:** Tatjana Seizova-Cajic, Janet L. Taylor.

**Data curation:** Tatjana Seizova-Cajic.

**Formal analysis:** Tatjana Seizova-Cajic.

**Investigation:** Sandra Ludvigsson, Birger Sourander, Melinda Popov.

**Methodology:** Sandra Ludvigsson, Birger Sourander, Janet L. Taylor.

**Project administration:** Tatjana Seizova-Cajic.

**Resources:** Tatjana Seizova-Cajic, Janet L. Taylor.

**Supervision:** Tatjana Seizova-Cajic.

**Visualization:** Tatjana Seizova-Cajic.

**Writing – original draft:** Tatjana Seizova-Cajic.

**Writing – review & editing:** Tatjana Seizova-Cajic, Janet L. Taylor.

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
