## [Decision Letter · Decision Letter 0]

19 Feb 2020

PONE-D-19-34498

Scrambling the skin: Simulated skin re-arrangement using apparent motion

PLOS ONE

Dear Dr. Seizova-Cajic,

Thank you for submitting your manuscript to PLOS ONE. After careful consideration, we feel that it has merit but does not fully meet PLOS ONE’s publication criteria as it currently stands. Therefore, we invite you to submit a revised version of the manuscript that addresses the points raised during the review process.

As you can see, the expert reviewers express significant concerns about the conclusions drawn from the data. I am frankly not convinced you can address them without additional experiments, but I am willing to give you the opportunity to try. 

We would appreciate receiving your revised manuscript by Apr 04 2020 11:59PM. To enhance the reproducibility of your results, we recommend that if applicable you deposit your laboratory protocols in protocols.io, where a protocol can be assigned its own identifier (DOI) such that it can be cited independently in the future. For instructions see: http://journals.plos.org/plosone/s/submission-guidelines#loc-laboratory-protocols

We look forward to receiving your revised manuscript.

Kind regards,

Sliman J. Bensmaia

Academic Editor

PLOS ONE

Journal Requirements:

Reviewers' comments:

Reviewer's Responses to Questions

**Comments to the Author**

1. Is the manuscript technically sound, and do the data support the conclusions?

Reviewer #1: Partly

Reviewer #2: Partly

2. Has the statistical analysis been performed appropriately and rigorously? 

Reviewer #1: No

Reviewer #2: Yes

3. Have the authors made all data underlying the findings in their manuscript fully available?

Reviewer #1: Yes

Reviewer #2: Yes

4. Is the manuscript presented in an intelligible fashion and written in standard English?

Reviewer #1: No

Reviewer #2: Yes

5. Review Comments to the Author

Reviewer #1: The authors reported a variant of motion after effect using six vibrotactile devices. They conducted psychophysics experiment examining the effect of two different adapting stimuli. One adapting stimulus is called as “orderly” condition, in which the adapting vibrotactile feedback is provided in a one-way direction. Another adapting stimulus is called as “scrambled” condition, in which the adapting vibrotactile feedback is presented in a spatially zig-zag fashion. Authors compared the d-prime of these two conditions with non-conditioning situation, and found that d’ was smaller when the adapting stimulus was “scrambled” rather than “orderly” adapting stimulus. The authors also reported the subjective feelings of adapting stimulus.

The authors reported an intriguing phenomenon, however, it was too early to draw a conclusion from the current result. For example, the authors started that the “scrambled” condition is as if it were simulating skin re-arrangment in the central nervous system. It is quite hard to generalize that the somatosensory mapping is corrected based on current psychophysical experiment. If the authors would argue the possibility, the authors should have conducted a series of psychophysical experiments before drawing the conclusion.

Especially, this reviewer would like to clarify the following items before accepting this manuscript.

(1) Without examining neural responses, the authors discuss one possible mechanisms of organization of sensory maps. Why could the authors lead to a discussion on the mechanisms of organization of sensory maps only from one psychophysics experiment? There was a possibility that the task in the "scrambled" condition was hard to answer for some reasons. Did the authors consider another possibility?

(2) Title of current draft was too general. Moreover, “scrambling the skin” is hard to grasp what the authors would like to argue in this manuscript. I would recommend authors to modify the title of this manuscript in a more succinct fashion.

(3) In the abstract, the authors argued that “we propose that the somatosensory system was beginning to ‘correct’…, but how did the authors decide the somatosensory system was to correct…. The somatosensory system may update, or being confused because of unnatural tactile stimulus condition. Why the authors could interpret that the somatosensory system “correct”? Explain.

—

I would argue to accept this manuscript before the items mentioned above is going to be accepted.

Minor points:

(4) The authors should have written the version of SPSS.

(5) How the authors determine the number of participants, and what was the effect size?

(6) The authors excluded 30 responses because the authors judged these 30 data points were outliers. How did the authors determine the criteria of outlier? Explain.

Reviewer #2: A central problem in neuroscience is how the spatial organization of receptors in the periphery is mapped into central neurons in the sensory cortex, and how this mapping is shaped by stimuli. In this study, the authors evaluated a century-old hypothesis that motion across the skin is important for the organization of spatial maps in the brain. To test this hypothesis, they run and experiment were participants were requested to discriminate the direction of apparent motion produced by the sequential activation of vibrating tactors. Before each test stimulus, a priming stimulus was delivered were the activation of two neighbours tactors was occasionally inverted (Scrambled) or not (Orderly). The Scrambled priming was associated with a reduced motion sensitivity. However, this did not produce a systematic bias in motion direction that would provide a stronger support to their hypothesis.

The experimental question is interesting. In general, the study is technically sound. I have major concerns on the type of stimuli used and on the very short duration of the priming stimuli. See below for detailed comments.

In the classical study by Merzenich and Jenkins, plastic changes in the cortical organization followed several months after the skin flap surgery. In (Rosselet et al 2008), the animal was stimulated for 7 h with a brushing stimulus. Instead, in the current study, the duration of each priming stimulus last only 66 seconds, and Scrambled stimuli were immediately followed by test stimuli were tactors were activated in a sequential order. It seems extremely unlikely that such short stimulus may produce a reorganization in the somatosensory cortex…

The second issue is the usage of apparent motion by sequential activation of vibrotactor. The simplest explanation is that participants might have perceived the Scrambled priming for what it is, i.e., a random activation of tactors, rather than adjust their spatial map according to the novel stimulus.

Altogether, the short duration of the priming stimulus and the use of apparent motion may explain why the authors did not find a reversal in motion perception, but only a reduced sensitivity. Instead, the conclusion that “the somatosensory system was beginning to correct reversed local motion” is not sufficiently supported by the data.

Minor issues

The authors used signal detection theory to analyse the responses in the discrimination task. This is the “standard” method for detection tasks. It is also correct using it for discrimination tasks; however, using a psychometric function is a more common choice in this second case (e.g., probability of perceiving proximal motion w.r.t. motion direction). The discriminability of the stimulus and motion bias could be analysed in term of JND and PSE, respectively. The analysis can be extended to the sample population by means of General Linear Mixed Model - see for example (Agresti, 2002; Bates, Mächler, Bolker, & Walker, 2015). In case of two levels of the predictor (proximal and distal direction) GLMM are also called “models for matched pairs”.

It is not clear why the authors pre-processed the data in R and used SPSS for statistical inference. R is an excellent language for model fitting! See for example (Knoblauch & Maloney, 2012) for fitting psychophysical data in R, including GLMM. (However: I am fine with SPSS if the authors prefer using it).

6. PLOS authors have the option to publish the peer review history of their article (what does this mean?). If published, this will include your full peer review and any attached files.

Reviewer #1: No

Reviewer #2: No

---

## [Decision Letter · Decision Letter 1]

29 Jun 2020

PONE-D-19-34498R1

Scrambling the skin: Simulated skin re-arrangement using apparent motion

PLOS ONE

Dear Dr. Seizova-Cajic,

Thank you for submitting your manuscript to PLOS ONE. After careful consideration, we feel that it has merit but does not fully meet PLOS ONE’s publication criteria as it currently stands. Therefore, we invite you to submit a revised version of the manuscript that addresses the points raised during the review process.

I apologize for the long turn around time, but the two original reviewers were split on the manuscript so I had to seek a third opinion, from a highly qualified investigator. As you can see, the new reviewer also expresses significant concerns about whether the results support the conclusions. I am happy to give you another shot at this, but it will require satisfying at least two of the three reviewers, so only resubmit if you think you can do that. 

We look forward to receiving your revised manuscript.

Kind regards,

Sliman J. Bensmaia

Academic Editor

PLOS ONE

Reviewers' comments:

Reviewer's Responses to Questions

**Comments to the Author**

1. If the authors have adequately addressed your comments raised in a previous round of review and you feel that this manuscript is now acceptable for publication, you may indicate that here to bypass the “Comments to the Author” section, enter your conflict of interest statement in the “Confidential to Editor” section, and submit your "Accept" recommendation.

Reviewer #1: All comments have been addressed

Reviewer #2: (No Response)

Reviewer #3: (No Response)

2. Is the manuscript technically sound, and do the data support the conclusions?

Reviewer #1: Yes

Reviewer #2: Partly

Reviewer #3: No

3. Has the statistical analysis been performed appropriately and rigorously? 

Reviewer #1: Yes

Reviewer #2: Yes

Reviewer #3: Yes

4. Have the authors made all data underlying the findings in their manuscript fully available?

Reviewer #1: Yes

Reviewer #2: Yes

Reviewer #3: Yes

5. Is the manuscript presented in an intelligible fashion and written in standard English?

Reviewer #1: Yes

Reviewer #2: Yes

Reviewer #3: Yes

6. Review Comments to the Author

Reviewer #1: I appreciated that the authors clarified the items that I had raised in the previous review process. I'm now getting comfortable with the authors' responses, but still, I would like to point out the following items.

(1) Thank you for the authors' efforts on additional statistical descriptions showing the effect size (such as confidence interval). I had looked at the data that the authors uploaded in the data repository (in SPSS format), and conducted some trials and errors before adopting the reported model as indicated in Table 2. The readers would benefit from the fact how the authors had chosen a quadratic model for the difference in sensitivity (d') between orderly and scrambled conditions from baseline, and a linear model for the difference in absolute bias. Also the authors had decided which model could describe the data the best based on information criteria, so it would be helpful for readers to conduct the model selection process in the data analysis section.

Minor comments:

(2) I'm still not very comfortable with the title. Even I would admit the catchy running title ("Scrambling the skin"), there is room to improve in the second part of the title. I initially imagined that the authors conducted a computational simulation of neural representation when I glanced at the title "simulated skin re-arrangement". Moreover, some of the readers may misinterpret "using apparent motion" as "using (visual) apparent motion", even though the authors addressed a question using tactile apparent motion. This is my personal opinion though.

(3) I did not notice in the first review process, however, I realized that the degrees of freedom of numerator and denominator in F distribution were not correctly indicated (p.12, F(78.2, 1) = 7.82 should be F(1, 78.2), and other similar four mistakes in the same paragraph).

Reviewer #2: In this study, the authors evaluated a century-old hypothesis that motion across the skin is important for the organization of spatial maps in the brain.

In the first review, both me and the other reviewer expressed major concerns about the experimental design, the analysis, and the conclusions of the study. In particular, I raised two major issues:

1. The very short duration of the conditioning stimuli (here the total duration of conditioning stimuli across the experiment is irrelevant because conditioning stimuli are alternated with "non-scrambled" control stimuli, restoring the putative mapping)

2. The usage of apparent "scrambled" motion instead of "scrambled" slip motion. Apparent motion does not reproduce important features of tactile motion, such as shear force and skin strain propagation.

I expected major changes in the manuscript; for example, the authors could address the first point with a simple control experiment using the same experimental setup, by exposing participants to a much prolonged conditioning stimulus. Instead, the authors only performed minor manuscript changes. The authors added that "Consistent changes across a number of spatial tasks would be strong evidence of structural changes" but they did not test for it.

For the reasons I explained before, I am not convinced that the short stimuli used in the study would produce the relevant plastic changes postulated by the authors. The electrophysiological studies now included in the manuscript refer to different phenomena, such as the change in the receptive field in cat's visual cortex, and not to the remapping of a skin portion in response to motion. The decrease in sensitivity reported by the authors could be due to a simple masking effect; instead, a bias change would provide much stronger evidence for remapping.

Overall, I am not convinced that the conclusions of the manuscript adequately supported by the present results.

Reviewer #3: In this manuscript, the authors study the effects of adaptation on motion perception. The authors use two different stimuli, an orderly and a scrambled stimulus, as adaptors, and find that performance is more strongly degraded for the scrambled adaptor. The experiments are well executed and the results and analysis are convincing. However, I share the serious reservations that Reviewers 1 and 2 brought up about the framing and interpretation of these results.

First, the authors claim in their Abstract that their results may represent “one possible mechanism for organization of sensory maps.” In their Conclusion, they claim that their experiments support “the idea that motion plays a major role in organizing spatial maps in touch,” and that “Our simulated-surgery paradigm is a potentially useful tool in the experimental study of plasticity in sensory maps in conscious humans.” However, the authors also explain that the results from this study are most likely mediated by short-term adaptation mechanisms, which are almost entirely distinct from the long-term remapping mechanisms that govern the organization of sensory maps.

To be more specific, adaptation is a short-term decrease in neural responsivity that occurs in response to an excess of incoming activity, and is thought to depend on gain-control mechanisms internal to individual cortical cells (c.f. Sanchez-Vives et. al. 2000). In contrast, remapping is likely a long-term homeostatic response to a lack of "feedforward" activity, and is associated with the strengthening of existing synaptic connections and growth of new neural connections. Because physiological literature suggests that these are two largely separate processes, it’s not clear to me that the results of adaptation psychophysics can be meaningfully applied to questions of remapping.

Second, the authors claim in the Abstract that if subjects were exposed to a much longer duration of adapting stimulation than tested in this study, they would expect “a complete reversal in perceived motion direction.” The data presented in this manuscript cannot distinguish between mechanisms that are weakening the sensitivities at sites 3 and 4 (that is, turning the neural mechanisms off) from mechanisms that swap the sensitivities of sites 3 and 4. I believe that the swapping result would be novel – I am not aware of any prior psychophysical studies in which the ordinal relationship of two spatial locations was reversed due to an extended period of adaptation. Additionally, adaptation effects at different timescales are likely mediated by different mechanisms (c.f. Solomon & Kohn 2014), so results at short timescales may not extrapolate to longer timescales. If the authors wish to talk about what happens when adapting long timescales, they must test longer timescales. As it stands, these data speak to the effects of short-term adaptation.

The authors helpfully point out that the higher average speed of the scrambled stimulus may lead to it being a more effective adaptor than the orderly stimulus. This explanation is well supported by previous work from the corresponding author, and seems to be more parsimonious than remapping.

7. PLOS authors have the option to publish the peer review history of their article (what does this mean?). If published, this will include your full peer review and any attached files.

Reviewer #1: No

Reviewer #2: No

Reviewer #3: No

---

## [Author Response · Author response to Decision Letter 1]

5 Aug 2020

Response to reviewers is attached.

---

## [Decision Letter · Decision Letter 2]

27 Oct 2020

PONE-D-19-34498R2

Scrambling the skin: Motion-scrambling paradigm for psychophysical study of somatosensory plasticity

PLOS ONE

Dear Dr. Seizova-Cajic,

Thank you for submitting your manuscript to PLOS ONE. After careful consideration, we feel that it has merit but does not fully meet PLOS ONE’s publication criteria as it currently stands. Therefore, we invite you to submit a revised version of the manuscript that addresses the points raised during the review process.

Unfortunately, the Scientific Editor, Dr. Sliman Bensmaia, is no longer available, so I took over the procedure.

The reviewers are all in unison, that the experimental data are interesting and really worth publishing.

However, they also all are uneasy with the interpretation overload that you give to the data. Naturally, it is important to give a theoretical background, but it is just as important not to overinterpret the results. Unless you refrain from such over-interpretation I am afraid we cannot publish your work, which would be a real loss considering the importance of your data.

Please keep in mind the comment by reviewer #1, to keep the text simple. As soon as you are having a hard time to explain to tell it in a simple way, it might well be that you are over-interpreting your data.

Please also follow the suggestions of reviewers #2 and #3 to change the title (#2, #3),i.e., to remove/replace “plasticity”. I encourage you to follow reviewer #3 to include “apparent motion” in the title, because that is how the paradigm is called (and well-known) in vision research. 

I also have some comments of my own:

Abstract: keep to the data and results of the present study, and so please remove the last sentence “Longer conditioning … without reversing.”Questions remain: – With respect to the baseline, the orderly stimulus is significantly deteriorated. How well is the baseline matched? Possibly better would be a baseline of vibration without motion direction, in order to remove the possible masking effect of stimulating the flanking locations. – How well is the “orderly” condition matched to the “scrambled” one? It would be good to have a reference condition that has an amount of “scrambling” in the flanking positions.– How is the relation of the distance between the stimulus locations to the receptive field size (e.g. two-point discrimination)?– The explanation heavily focuses on spatial remapping, but an alternative explanation would be temporal remapping. 

These remaining questions call for further experiments, as you have yourselves agreed. Such experiments need not be included in the present study, but they further stress the preliminary nature of your results. 

These questions need to be addressed briefly in the discussion, and you need to refrain from over-interpreting your results. 

We look forward to receiving your revised manuscript.

Good Luck!

Kind regards,

Marc H.E. de Lussanet, Ph.D.

Academic Editor

PLOS ONE

Reviewers' comments:

Reviewer's Responses to Questions

**Comments to the Author**

1. If the authors have adequately addressed your comments raised in a previous round of review and you feel that this manuscript is now acceptable for publication, you may indicate that here to bypass the “Comments to the Author” section, enter your conflict of interest statement in the “Confidential to Editor” section, and submit your "Accept" recommendation.

Reviewer #1: All comments have been addressed

Reviewer #2: (No Response)

Reviewer #3: (No Response)

2. Is the manuscript technically sound, and do the data support the conclusions?

Reviewer #1: Partly

Reviewer #2: Yes

Reviewer #3: Yes

3. Has the statistical analysis been performed appropriately and rigorously? 

Reviewer #1: Yes

Reviewer #2: Yes

Reviewer #3: Yes

4. Have the authors made all data underlying the findings in their manuscript fully available?

Reviewer #1: Yes

Reviewer #2: Yes

Reviewer #3: Yes

5. Is the manuscript presented in an intelligible fashion and written in standard English?

Reviewer #1: Yes

Reviewer #2: Yes

Reviewer #3: Yes

6. Review Comments to the Author

Reviewer #1: Thank you very much for great efforts on revising this manuscript. By this revision, the authors clearly show their attitude that main objective of this study is not to show the psychological phenomenon (degraded motion discrimination after the exposure to scrambled pattern), but to show as an evidence of somatosensory plasticity after conditioning stimulus for 66 seconds (and repeated trials for 24 times). In addition, the authors updated the abstract and stated that "The 26-minutes conditioning The 26-minutes conditioning was interspersed with test motion between the two middle tactors presented on their own (sequence 3-4 or 4-3)" even though the actual duration of conditioning stimulus was 66 seconds for each test cycle. This is misleading change and it appears that the authors tried to overstate the observed psychological phenomenon.

I fully appreciated that the authors added detail information on statistical analysis procedures and criteria. It helps readers to follow the logic in the data analysis. However, I would like to change my previous decision if the authors would show their data as an evidence of somatosensory plasticity. In current manuscript, the argument is focused on this hypothetical statement, rather not reporting the phenomenon. The text is getting complicated and hard to follow the main psychological result, which was worth publishing. If the authors could modify the manuscript by simply reporting the observed phenomenon, there would be a chance to be published.

Reviewer #2: The authors addressed my previous comments. I still have some reservations about the interpretation of the results. I suppose, however, the study can be published, so that the hypothesis can be further tested and refined. I only have two minor comments that can be managed during the editorial process:

I suggest to remove from the title the word “plasticity”, which would be misleading for the readers and replace for e.g. with “re-learning of relative positions” or similar.

Please expand the Table 1 caption. I am not sure about the meaning of column labels Proximal/Distal (stimulus motion direction?) and Elbow to wrist/Wrist to elbow.

Reviewer #3: In my opinion, the new version of the manuscript is much improved. The authors have done a good job of qualifying their original claims of plasticity, and now appropriately consider adaptation as an alternative explanation. I am satisfied outside of a few, relatively minor concerns.

First, I continue to be uncomfortable with the word “Plasticity” in the title of the paper. I understand that the method will eventually be used in a plasticity paradigm, but the data in this paper speak to a shorter timescale process. Any link to plasticity is, at this point, purely speculative. Terms such as “Adaptation” or “Changes in Somatosensory Sensitivity” would be more appropriate.

Second, I remain unconvinced by the idea proposed in the Discussion that “frequent direction change” would necessarily lead to weaker motion adaptation than a steady, unchanging motion stimulus. In practice, the short-term adaptation properties (occurring in the first ~100 ms of the response) of neurons in both the somatosensory and visual systems mean that they are often more strongly driven by transient changes in stimulus strength than by steady stimulation (e.g. Connor and Johnson 1990, Lisberger and Movshon 1999). As such, my prediction would run opposite that of the authors, in that I would expect rapidly changing motion stimuli to be a stronger adaptor than consistent motion of the same average speed.

The authors are entitled to their point of view. However, the “frequent direction change” paragraph, as it stands, is mere conjecture. Its claim is much less well supported than the counterbalancing claim that higher speeds lead to stronger adaptation. This point would be strengthened by either references or a more developed argument that supports the idea that rapidly changing motion stimuli are less effective than consistent motion stimuli at driving adaptation.

Please note that the cited text in the replies to reviewers (beginning “In brief, higher speed…”) is not present in the current version of the paper.

7. PLOS authors have the option to publish the peer review history of their article (what does this mean?). If published, this will include your full peer review and any attached files.

Reviewer #1: No

Reviewer #2: No

Reviewer #3: No

---

## [Author Response · Author response to Decision Letter 2]

11 Nov 2020

Reply to reviewers is attached as a separate file.

---

## [Editor Report · Decision Letter 3]

25 Nov 2020

PONE-D-19-34498R3

Scrambling the skin: A psychophysical study of adaptation to scrambled apparent motion

PLOS ONE

Dear Dr. Seizova-Cajic,

Thank you for submitting your manuscript to PLOS ONE. After careful consideration, we invite you to submit a revised version of the manuscript.

One minor issue remains. In response to the comments to the title, you have changed it back to the original title of the first submission. However, the reason why you changed that original title was because it was confusing, as you acknowledge. Therefore, please change the title again to make it more clear. For example, by adding one or more words such as: "tactile apparent motion" or: "apparent motion from tactors on the skin". 

As a final note: From reading the comments, I do not have the impression that the reviewers have a different opinion "about what a scientific paper should include". The three experts wer in unison that you promoted an interpretation that is not supported by the data, and that is not in agreement with the guidelines of PLoS ONE. The changes that you made (for example in the last sentence of the abstract and in the Discussion) now make clear that different interpretations are possible and on the basis of the current experiment no definite conclusion is possible. That is fine. Congratulations!

We look forward to receiving your revised manuscript.

Kind regards,

Marc H.E. de Lussanet, Ph.D.

Academic Editor

PLOS ONE

---

## [Editor Report · Decision Letter 4]

1 Dec 2020

Scrambling the skin: A psychophysical study of adaptation to scrambled tactile apparent motion

PONE-D-19-34498R4

Dear Dr. Seizova-Cajic,

We’re really pleased to inform you that your manuscript has been judged scientifically suitable for publication and will be formally accepted for publication once it meets all outstanding technical requirements.

Kind regards,

Marc H.E. de Lussanet, Ph.D.

Academic Editor

PLOS ONE
---

## [Editor Report · Acceptance letter]

22 Dec 2020

PONE-D-19-34498R4 

Scrambling the skin: A psychophysical study of adaptation to scrambled tactile apparent motion 

Dear Dr. Seizova-Cajic:

I'm pleased to inform you that your manuscript has been deemed suitable for publication in PLOS ONE. Congratulations! Your manuscript is now with our production department. 

Kind regards, 

on behalf of

Dr. Marc H.E. de Lussanet 

Academic Editor

PLOS ONE